# The Chimeric Peptide (GEP44) Reduces Body Weight and Both Energy Intake and Energy Expenditure in Diet-Induced Obese Rats

**DOI:** 10.3390/ijms26073032

**Published:** 2025-03-26

**Authors:** Matvey Goldberg, James E. Blevins, Tami Wolden-Hanson, Clinton T. Elfers, Kylie S. Chichura, Emily F. Ashlaw, Laura J. den Hartigh, Christian L. Roth, Robert P. Doyle

**Affiliations:** 1VA Puget Sound Health Care System, Office of Research and Development Medical Research Service, Department of Veterans Affairs Medical Center, Seattle, WA 98108, USA; matveyg@uw.edu (M.G.); tami.woldenhanson@gmail.com (T.W.-H.); 2Division of Metabolism, Endocrinology and Nutrition, Department of Medicine, University of Washington School of Medicine, Seattle, WA 98195, USA; lauradh@uw.edu; 3Seattle Children’s Research Institute, Seattle, WA 98195, USA; clinton.elfers@seattlechildrens.org (C.T.E.); christian.roth@seattlechildrens.org (C.L.R.); 4Department of Chemistry, Syracuse University, Syracuse, NY 13244, USA; kchichura@alltrna.com (K.S.C.); efashlaw@syr.edu (E.F.A.); rpdoyle@syr.edu (R.P.D.); 5UW Medicine Diabetes Institute, University of Washington School of Medicine, Seattle, WA 98109, USA; 6Department of Pediatrics, University of Washington School of Medicine, Seattle, WA 98195, USA; 7Departments of Medicine and Pharmacology, SUNY Upstate Medical University, Syracuse, NY 13244, USA

**Keywords:** obesity, multi-agonist, GLP-1, PYY, iWAT, iBAT

## Abstract

We recently reported that a chimeric peptide (GEP44) targeting the glucagon-like peptide-1 receptor (GLP-1R) and neuropeptide Y1- and Y2- receptors decreased body weight (BW), energy intake, and core temperature in diet-induced obese (DIO) male and female mice. In the current study, we tested the hypothesis that the strong reduction in body weight in response to GEP44 is partially related to the stimulation of energy expenditure (EE). To test this, rats were maintained on a high fat diet (HFD) for at least 4 months to elicit DIO prior to undergoing a sequential 2-day vehicle period, 2-day GEP44 (50 nmol/kg) period, and a minimum 2-day washout period, and detailed measures of energy homeostasis. GEP44 (50 nmol/kg) reduced EE (indirect calorimetry), respiratory exchange ratio (RER), core temperature, activity, energy intake, and BW in male and female rats. As in our previous study in mice, GEP44 reduced BW in male and female HFD-fed rats by 3.8 ± 0.2% and 2.3 ± 0.4%, respectively. These effects appear to be mediated by increased lipid oxidation and reductions in energy intake as GEP44 reduced RER and cumulative energy intake in male and female HFD-fed rats. The strong reduction in body weight in response to GEP44 is related to a robust reduction in energy intake, but not to the stimulation of EE. The paradoxical finding that GEP44 reduced EE might be secondary to a reduction in diet-induced thermogenesis or might indicate an important mechanism to limit the overall efficacy of GEP44 to prevent further weight loss.

## 1. Introduction

The control of body weight involves a complex interaction of factors that control food intake and energy expenditure (EE). While many risk factors (i.e., dietary, environmental, genetic, and psychological) are thought to contribute to obesity, increased energy intake without corresponding increases in energy output (EE, activity) contribute, in part, to weight gain and subsequent obesity [1]. The obesity epidemic and its associated complications, including cardiovascular disease, type 2 diabetes, osteoarthritis, depression, certain types of cancer, and COVID-19-related hospitalizations, are global health concerns. More than one billion people in the world are obese [2]. Up to 50% of adults in the US alone are predicted to become obese by 2030 [3]. Furthermore, the costs to treat obesity in the US by 2030 are estimated to be an alarming 3 trillion/year [4]. Once-weekly treatment with the selective glucagon-like peptide-1 receptor (GLP-1R) agoniost, semaglutide, has resulted in a prolonged loss in body weight that has ranged from ≈6.7% over 40 weeks [5], ≈14.9% over 68 weeks [6], and ≈10.2% over 208 weeks [7], further highlighting the need to optimize anti-obesity treatment options that result in more marked weight loss over more sustained treatment intervals. In addition, there is rapid weight regain when such treatment options are discontinued [8]. The limitation of pharmacotherapies to promote long-term weight loss (>10%) in humans who are overweight or obese is thought to occur, in part, by activation of counter-regulatory orexigenic mechanisms in the CNS that drive hunger and energy conservation mechanisms that reduce EE, thus preventing further weight loss and promoting weight regain [9].

Recent findings indicate that combination therapy (co-administration of different compounds) or treatment with monomeric dual- and triple-agonists at low-dose or subthreshold dose combinations may be more effective than monotherapy for producing sustained weight loss [10]. The increased efficacy of these compounds is likely due, in part, to their ability to both suppress energy intake and/or increase EE, as well as to prevent the counter-regulatory orexigenic effects mentioned above. One example of a dual-agonist is Tirzepatide (Zepbound^TM^), which targets both glucose-dependent insulinotropic polypeptide receptor (GIPR) and GLP-1R and was approved by the FDA in November 2023 for weight management. It has been found to elicit more profound weight loss in humans when compared to semaglutide monotherapy. Prolonged treatment over a 72- [11] and 88-week [12] period has resulted in a profound weight loss of 20.9% and 25.3%, respectively. More recently, retatrutide, a triple-agonist that targets GLP-1R, GIPR, and glucagon receptors (GCGR), was able to elicit a sustained reduction in body weight by 24.2% over a 48-week period [13]. While these compounds evoke profound weight loss in the absence of bariatric surgery, they are still associated with a number of mild to moderate adverse gastrointestinal side effects, namely diarrhea, nausea, vomiting, and/or abdominal pain [11]. This stresses the need to develop new, effective, and safe strategies to treat the growing obesity epidemic.

We recently developed the triple agonist, GEP44, which is a partial agonist at GLP-1R, Y1R, and Y2R [14]. GEP44 was originally designed to treat type 2 diabetes (T2DM) and obesity. We recently found that GEP44 reduced energy intake, body weight, core temperature, and glucose levels in male and female DIO mice primarily through a GLP-1R-dependent mechanism. However, the remaining open question was whether the reduction in food intake is accompanied by a reduction in EE (using indirect calorimetry), which can limit the overall success of sustained weight loss. In the current study, we tested the hypothesis that the strong reduction in body weight in response to GEP44 is partially related to the stimulation of energy expenditure (EE) in addition to a reduction in energy intake. We also measured gross motor activity to test if spontaneous physical activity [15] or shivering and non-shivering thermogenesis in skeletal muscle [16] might contribute to GEP44-elicited reductions in EE. The selective GLP-1R agonist, exendin-4, was included as a control to assess GLP-1R-mediated effects on EE [17].

## 2. Results

### 2.1. Study 1.1: Determine the Effects of GEP44 and Exendin-4 on Signs of Pain or Discomfort, Body Weight, and Energy Intake in Male and Female HFD-Fed Rats

The overall goal of these studies was to determine the extent to which the chimeric peptide, GEP44, produces signs of pain or discomfort and impacts core temperature (as a surrogate for EE) and activity, and EE at doses that reduce body weight and energy intake in both male DIO and female HFD-fed rats. In addition, we incorporated the use of the selective GLP-1R agonist, exendin-4, to assess GLP-1R-mediated effects on body weight, energy intake, core temperature, activity, and EE in male and female HFD-fed rats.

### 2.2. Observational Signs of Pain or Discomfort

The number of animals that displayed signs of pain or discomfort (elongated or flattened nose, narrow eyes) as described in [18] is shown in Table 1.

### 2.3. GEP44 (Males)

As previously shown [14], there appeared to be increased signs of pain or discomfort following GLP-1R agonist treatment. Male rats treated with GEP44 showed a modest increase in the signs of discomfort, including elongated or flattened noses (*p* < 0.05) and narrow eyes (*p* < 0.05) relative to vehicle treatment. There were five GEP44-treated rats displaying such signs over 2 treatment days, compared to zero in vehicle-treated rats.

### 2.4. Exendin-4 (Males)

Compared to GEP44, exendin-4 treatment resulted in significantly more incidents of these elongated or flattened noses (*p* < 0.01) or narrow eyes (*p* < 0.01). There were 23 exendin-4-treated male rats showing such signs over 2 treatment days, compared to 0 in vehicle-treated rats.

### 2.5. GEP44 (Females)

Female rats treated with GEP44 showed a modest increase in the signs of discomfort, including elongated or flattened noses (*p* < 0.05) and narrow eyes (*p* < 0.05) relative to vehicle treatment. There were eight rats displaying such signs over 2 treatment days, compared to zero in vehicle-treated rats.

### 2.6. Exendin-4 (Females)

Female rats treated with exendin-4 showed an increase in the signs of discomfort, including elongated or flattened noses (*p* < 0.01) and narrow eyes (*p* < 0.01) relative to vehicle treatment. There were 13 rats showing such signs over 2 treatment days, compared to 0 in vehicle-treated rats.

### 2.7. Body Weight

Both male and female Wistar rats were weight-matched within respective groups prior to treatment onset. GEP44 treatment reduced cumulative 2-day body weight in both male DIO (Figure 1A; *p* < 0.05) and female HFD-fed rats (Figure 1A; *p* < 0.05). Specifically, GEP44 reduced body weight after a 2-day treatment in male DIO and female HFD-fed rats by −29.3 ± 1.7 and −8.3 ± 1.5 g, respectively (*p* < 0.05). This amounted to approximately 3.8 ± 0.2% and 2.3 ± 0.4% weight loss in male and female rats, respectively.

Similarly, the selective GLP-1R agonist, exendin-4, reduced body weight in both male (Figure 1B; *p* < 0.05) and female HFD-fed rats (Figure 1B; *p* < 0.05). Specifically, exendin-4 reduced body weight after a 2-day treatment in male and female HFD-fed rats by −26.5 ± 1.7 and −7.6 ± 1.4 g, respectively (*p* < 0.05). This amounted to approximately 3.4 ± 0.2% and 2.2 ± 0.4% weight loss in male and female rats, respectively. Note that exendin-4 was administered at a slightly lower dose (5 nmol/kg instead of 10 nmol/kg) on treatment day 2. However, there was no difference in 24-h energy intake in response to exendin-4 treatment at 5 (41.9 ± 5.0 kcal) and 10 nmol/kg from either the same animals (43.8 ± 4.1 kcal; *p* = NS) or a different subset of animals (44.2 ± 3.5 kcal; *p* = NS). Therefore, we combined the data together as part of the analysis with the animals that received exendin-4 at 10 nmol/kg.

### 2.8. Effect of GEP44 vs. Exendin-4 on Body Weight in Male and Female Rats

There was no significant difference in the effectiveness of GEP44 and exendin-4 in reducing total body weight at 48-h post-injection in male or female rats (*p* = NS). This was the case whether the data were analyzed by absolute weight loss (grams) or % weight loss.

### 2.9. Energy Intake

GEP44 treatment significantly reduced cumulative energy intake following treatment days 1 and 2 in both male (Figure 1C; *p* < 0.05) and female rats (Figure 1C; *p* < 0.05). In males, GEP44 reduced cumulative 24 and 48 h energy intake by approximately 76.6 ± 3.1% and 74.6 ± 2.3%, respectively. In females, GEP44 reduced cumulative 24 and 48-h energy intake by approximately 57.4 ± 3.0% and 63 ± 2.4% suppression, respectively.

Similarly, exendin-4 treatment also significantly reduced cumulative energy intake following treatment days 1 and 2 in both male (Figure 1D; *p* < 0.05) and female rats (Figure 1D; *p* < 0.05). In males, exendin-4 reduced cumulative 24 and 48 h energy intake by approximately 70.8 ± 2.8% and 70.0 ± 2.8%, respectively. In females, exendin-4 reduced cumulative 24 and 48-h energy intake by approximately 68.5 ± 3.6% and 69 ± 2.4% suppression, respectively.

### 2.10. Effect of GEP44 vs. Exendin-4 on Energy Intake in Male and Female Rats

GEP44 was more effective than exendin-4 at reducing cumulative energy intake over 24 [F(1,18) = 4.848, *p* = 0.041] and 48-h post-injection [F(1,18) = 14.005, *p* = 0.001] in male rats. On the other hand, exendin-4 was more effective than GEP44 at reducing cumulative energy intake over 24 [F(1,21) = 13.586, *p* = 0.001] and 48-h post-injection [F(1,21) = 12.375, *p* = 0.002].

### 2.11. Study 1.2A-B: Determine the Effects of GEP44 and Exendin-4 on (A) Core Temperature and (B) Gross Motor Activity in Male and Female HFD-Fed Rats

#### 2.11.1. Study 1.2A-B: Changes in (A) Core Temperature and (B) Gross Motor Activity

The analysis focused on the effects of GEP44 and exendin-4 on core temperature and activity at 6-h post-injection (dark cycle) and 23-h post-injection period. We also analyzed the extent to which these effects carried into the 12 h dark cycle and 11 h light cycle. Note that the last hour of the light cycle (hour 12) was excluded from the core temperature and activity analysis as this was during the time that the animals were being handled and injected. The core temperature and gross motor activity data represent data that were averaged over 6 h, 12 h dark cycle, 11 h light cycle, and 23-h post-injection period.

#### 2.11.2. Study 1.2A: Core Temperature (Males)

GEP44 and exendin-4 both reduced core temperature throughout the post-treatment period. GEP44 reduced core temperature over the 6-h post-injection (Figure 2A; *p* < 0.05) and 23-h post-injection period in both male rats (Figure 2B; *p* < 0.05). Notably, similar results were observed during the 12 h dark cycle in response to GEP44 (*p* < 0.05; Appendix A). GEP44 also elicited a slight elevation of core temp during the 11 h light period (*p* < 0.05; Appendix A).

Exendin-4 also reduced core temperature over the 6-h post-injection (Figure 2C; *p* < 0.05) and 23-h post-injection period in male rats (Figure 2D; *p* < 0.05). Notably, similar results were observed during the 12 h dark cycle in response to exendin-4 (*p* < 0.05; Appendix A). Like GEP44, exendin-4 also elicited a slight elevation of core temp during the 11 h light period (*p* < 0.05; Appendix A).

#### 2.11.3. Core Temperature (Females)

GEP44 and exendin-4 both reduced core temperature throughout the post-treatment period. GEP44 reduced core temperature over the 6-h post-injection (Figure 2A; *p* < 0.05) and 23-h post-injection period in female rats (Figure 2B; *p* < 0.05). Notably, similar results were observed during the 12 h dark cycle in response to GEP44 (*p* < 0.05). GEP44 also elicited a slight elevation of core temp during the 11 h light period in female rats (*p* < 0.05).

Exendin-4 also reduced core temperature over the 6-h post-injection (Figure 2C; *p* < 0.05) and 23-h post-injection period in female rats (Figure 2D; *p* < 0.05). Notably, similar results were observed during the 12 h dark cycle in response to exendin-4 in female rats (*p* < 0.05). Like GEP44, exendin-4 also elicited a slight elevation of core temp during the 11 h light period (*p* < 0.05).

### 2.12. Effect of GEP44 vs. Exendin-4 on Core Temperature in Male and Female Rats

GEP44 was more effective than exendin-4 at reducing average core temperature over 6 h [F(1,14) = 7.685, *p* = 0.015], the dark cycle [F(1,14) = 25.173, *p* < 0.01], and 23-h post-injection [F(1,14) = 9.809, *p* = 0.007]. GEP44 also tended to be more effective at reducing core temperature during the light cycle [F(1,14) = 3.631, *p* = 0.077] in male rats.

GEP44 was also more effective than exendin-4 at reducing average core temperature over 6 h [F(1,6) = 15.570, *p* = 0.008], the dark cycle [F(1,6) = 10.696, *p* = 0.017], and 23 h [F(1,6) = 6.828, *p* = 0.04] in female rats.

### 2.13. Study 1.2B: Gross Motor Activity (Males)

GEP44 and exendin-4 both reduced gross motor activity throughout the post-treatment period. GEP44 reduced gross motor activity over the 6-h post-injection (Appendix A; *p* < 0.05) and 23-h post-injection period in male rats (Appendix A; *p* < 0.05). Notably, similar results were observed during the 12 h dark cycle in response to GEP44 in male rats (*p* < 0.05; Appendix A). GEP44 failed to significantly impact activity during the 11 h light cycle (*p* = NS; Appendix A).

Exendin-4 also reduced gross motor activity over the 6-h post-injection (Appendix A; *p* < 0.05) and 23-h post-injection period in male rats (Appendix A; *p* < 0.05). Notably, similar results were observed during the 12 h dark cycle in response to exendin-4 in male rats (*p* < 0.05; Appendix A). Exendin also tended to reduce activity during the 11 h light cycle, but this was not significant (*p* = 0.114; Appendix A).

### 2.14. Effect of GEP44 vs. Exendin-4 on Gross Motor Activity in Male Rats

There were no differences in the effectiveness of GEP44 and exendin-4 on gross motor activity over 6 h, the light cycle, or 23-h post-injection in male rats. GEP44 tended to produce a more profound reduction in activity during the dark cycle [F(1,14) = 2.511, *p* = 0.135] when compared to exendin-4 in male rats.

### 2.15. Sex Differences Linked to Effects of GEP44 and Exendin-4 on Body Weight, Energy Intake, and Core Temperature

#### 2.15.1. GEP44

We ran a two-way ANOVA in order to examine sex differences with respect to drug treatment, which resulted in a near-significant overall effect of sex [F(1,78) = 3.350, *p* = 0.071] and a significant interactive effect of GEP44 and sex to elicit weight loss [F(1,78) = 10.779, *p* < 0.01].

Specifically, there was a significant difference in the effectiveness of GEP44 (50 nmol/kg) to reduce cumulative body weight over the 48-h treatment period with there being a more pronounced effect of GEP44 to reduce body weight in males compared to females [F(1,39) = 10.873, *p* = 0.002].

Two-way ANOVA also demonstrated a significant overall effect of sex [F(1,78) = 49.391, *p* < 0.01] and a significant interactive effect of GEP44 and sex to reduce 24-h energy intake [F(1,78) = 55.582, *p* < 0.01]. Similarly, there was also a significant overall effect of GEP44 to reduce 48-h energy intake in male and female rats [F(1,78) = 597.596, *p* < 0.01], a significant overall effect of sex [F(1,78) = 91.757, *p* < 0.01], and a significant interactive effect of GEP44 and sex to reduce 48-h energy intake [F(1,78) = 65.707, *p* < 0.01].

While the energy intake was not significantly different in response to GEP44 between male and female rats (possibly due to a floor effect on energy intake), males ate more than females at 24 [F(1,39) = 81.152, *p* < 0.01] and 48 h [F(1,39) = 113.861, *p* < 0.01] following vehicle administration. This contributed to a more pronounced suppression of energy intake in response to GEP44 at 24 [F(1,39) = 19.721, *p* < 0.01] and 48 h [F(1,39) = 12.118, *p* = 0.001] post-injection in males relative to females. Collectively, these findings indicate that there was a greater effectiveness of GEP44 in reducing body weight and cumulative 24 and 48-h energy intake in male compared to female rats.

We also found a more pronounced reduction in core temperature in the absence of any drug treatment (following vehicle treatment) in male rats relative to female rats. Male rats showed a more pronounced reduction in core temperature during the 6-h post-vehicle injection period [F(1,20) = 6.918, *p* = 0.016], 12 h dark cycle [F(1,20) = 6.482, *p* = 0.019], 11 h light cycle [F(1,20) = 10.979, *p* = 0.003], and 23-h [F(1,20) = 8.566, *p* = 0.008] post-vehicle injection period. Similarly, we found a reduction in core temperature in male rats independent of any treatment during baseline measurements (in the absence of vehicle treatment) during 6 h [F(1,21) = 13.572, *p* = 0.001], 12 h dark [F(1,21) = 11.037, *p* = 0.003], 11 h light [F(1,21) = 13.081, *p* = 0.002], and 23-h post-injection periods [F(1,21) = 11.131, *p* = 0.003] following the start of the dark cycle on the day prior to vehicle treatment.

Two-way ANOVA demonstrated a significant overall effect of GEP44 to reduce 6-h core temperature in male and female rats [F(1,40) = 134.097, *p* < 0.01] and a significant overall effect of sex [F(1,40) = 6.168, *p* = 0.017], but there was no significant interactive effect of GEP44 and sex to reduce core temperature [F(1,40) = 0.004, *p* = 0.951]. There was also a significant overall effect of GEP44 to reduce core temperature during the dark cycle in male and female rats [F(1,40) = 52.228, *p* < 0.01], a significant overall effect of sex [F(1,40) = 13.584, *p* < 0.01], and a near-significant interactive effect of GEP44 and sex to reduce core temperature [F(1,40) = 2.050, *p* = 0.160]. Similar results were found with respect to the effects of GEP44 on core temperature during the light cycle and over the 23-h post-injection period, but there was no interactive effect.

Specifically, there was also a significant difference in the effectiveness of GEP44 in reducing core temperature during the dark cycle [F(1,20) = 8.251, *p* = 0.009], increasing core temperature during the light cycle [F(1,20) = 6.241, *p* = 0.021], and reducing core temperature over the 23-h post-injection period [F(1,20) = 8.012, *p* = 0.010], with males showing a more pronounced effect.

#### 2.15.2. Exendin-4

Two-way ANOVA did not demonstrate an overall effect of sex [F(1,78) = 0.015, *p* = 0.904] but there was an interactive effect of exendin-4 and sex to elicit weight loss [F(1,78) = 18.475, *p* < 0.01].

Specifically, there was a significant difference in the effectiveness of exendin-4 (10 nmol/kg) to elicit weight loss [F(1,39) = 6.793, *p* = 0.013] in male and female rats, with males showing a more heightened effect.

Two-way ANOVA demonstrated a significant overall effect of sex [F(1,78) = 135.213, *p* = <0.01] and an interactive effect of exendin-4 and sex to reduce 24-h cumulative energy intake [F(1,78) = 43.937, *p* < 0.01] in male and female rats. Similarly, there was also a significant overall effect of exendin-4 to reduce 48-h cumulative energy intake [F(1,78) = 582.163, *p* < 0.01], an overall effect of sex [F(1,78) = 193.789, *p* < 0.01], and an interactive effect of exendin-4 and sex to reduce 48-h energy intake in male and female rats [F(1,78) = 47.493, *p* < 0.01].

Specifically, this interactive effect on energy intake appears to be driven, in part, by a more pronounced reduction in energy intake (regardless of vehicle or exendin-4 treatment) in female rats. Similarly to GEP44 treatment, males ate more than females in response to exendin-4 at both 24 [F(1,39) = 14.098, *p* = 0.001] and 48-h post-injection [F(1,39) = 40.231, *p* < 0.01]. Similarly, males also ate more than females at 24 h [F(1,39) = 149.665, *p* < 0.01] and 48-h post-vehicle injection [F(1,39) = 156.271, *p* < 0.01]. In contrast to GEP44 treatment, there was no difference in the suppression of food intake between males and females at 24 h [F(1,39) = 0.231, *p* = 0.633] and 48-h post-exendin-4 injection [F(1,39) = 0.284, *p* = 0.597].

Collectively, these findings indicate that there appeared to be a greater effect of exendin-4 on weight loss in males and there was a more robust increase in energy intake (regardless of treatment) at both 24 and 48-h post-vehicle or exendin-4 treatment in males compared to females.

Two-way ANOVA demonstrated a significant overall effect of sex [F(1,40) = 10.118, *p* < 0.05] but no overall interactive effect of exendin-4 and sex to reduce 6-h core temperature [F(1,40) = 0.020, *p* = 0.887] in male and female rats. There was also a significant overall effect of exendin-4 to reduce dark cycle core temperature [F(1,40) = 30.380, *p* < 0.01], an overall effect of sex [F(1,40) = 15.340, *p* < 0.01] but no overall interactive effect of exendin-4 and sex to reduce dark cycle core temperature [F(1,40) = 0.639, *p* = 0.429] in male and female rats. Similarly, there was a significant overall effect of exendin-4 to reduce light cycle core temperature [F(1,40) = 12.115, *p* < 0.01], an overall effect of sex [F(1,40) = 21.610, *p* < 0.01] but no overall interactive effect of exendin-4 and sex to reduce light cycle core temperature [F(1,40) = 0.046, *p* = 0.832] in male and female rats. Lastly, there was a close to significant overall effect of exendin-4 to reduce 23-h core temperature [F(1,40) = 3.586, *p* = 0.066], an overall effect of sex [F(1,40) = 21.759, *p* < 0.01] but no overall interactive effect of exendin-4 and sex to reduce 23-h core temperature [F(1,40) = 0.267, *p* = 0.608] in male and female rats.

Specifically, there was a near-significant difference in the effectiveness of exendin-4 in reducing the core temperature at 6-h post-injection [F(1,20) = 4.144, *p* = 0.055]. This reached significance during the dark cycle [F(1,20) = 8.893, *p* = 0.007] and over the 23-h post-injection period [F(1,20) = 13.481, *p* = 0.002] with males showing a more pronounced effect. While exendin-4 treatment was associated with a slight elevation of core temperature during the light cycle in both males and females, exendin-4 treated males tended to have lower core temperatures during this time period [F(1,20) = 10.652, *p* = 0.004]. Collectively, these findings indicate that there was greater effectiveness of exendin-4 to impact core temperature during the dark cycle, light cycle, and 23-h post-injection measurement period in male compared to female rats.

### 2.16. Study 2.1A-C: Determine the Effects of GEP44 and Exendin-4 on Ambulatory Activity, EE, and RER in Male and Female HFD-Fed Rats

#### 2.16.1. Study 2.1A: Ambulatory Activity (Males)

We also measured ambulatory activity as a complementary strategy to our earlier measurements of gross motor activity that were obtained from the same animals in Study 1.2B. In addition, we assessed both EE and RER to further delineate the mechanisms underlying the anti-obesity effects of both GEP44 and exendin-4 and to complement our earlier measurements on core temperature in Study 3.

#### 2.16.2. GEP44 and Exendin-4 Both Reduced Activity Throughout the Post-Treatment Period

Consistent with our observations on gross motor activity during the dark cycle in male rats, GEP44 also reduced ambulatory activity over the dark cycle [F(1,15) = 43.364, *p* < 0.01]. In addition, GEP44 also reduced activity during the light cycle [F(1,15) = 15.293, *p* < 0.01] in male rats (Figure 3A; *p* < 0.05).

Similarly, exendin-4 also reduced activity over the dark cycle [F(1,15) = 39.856, *p* < 0.01]. In addition, exendin-4 also reduced activity during the light cycle [F(1,15) = 7.265, *p* < 0.05] in male rats (Figure 3C).

#### 2.16.3. Ambulatory Activity (Females)

GEP44 and exendin-4 both reduced activity throughout the post-treatment period. GEP44 reduced activity over the dark cycle [F(1,15) = 21.168, *p* < 0.01] but failed to impact ambulatory activity during the light cycle in female rats (*p* = 0.167; Figure 3B).

Exendin-4 also reduced activity over the dark cycle [F(1,15) = 18.893, *p* < 0.01] but failed to significantly impact activity during the light cycle in female rats (*p* = 0.341; Figure 3D).

### 2.17. Effect of GEP44 vs. Exendin-4 on Ambulatory Activity in Male and Female Rats

GEP44 was more effective than exendin-4 at reducing ambulatory activity over the light cycle [F(1,15) = 6.446, *p* = 0.023] in male rats but there were no differences between GEP44 and exendin-4 on ambulatory activity during the dark cycle. In contrast to male rats, there were no differences between GEP44 and exendin-4 on ambulatory activity over light or dark cycles in female rats.

### 2.18. Study 2.1B: EE (Males)

Consistent with our finding that GEP44 and exendin-4 both reduced core temperature in male rats (a surrogate marker of EE), GEP44 and exendin-4 also reduced EE throughout the post-treatment period. GEP44 reduced EE over the dark cycle [F(1,15) = 64.902, *p* < 0.01] in male rats (Figure 4A; *p* < 0.05). In contrast, GEP44 treatment was associated with an increase in EE during the light cycle [F(1,15) = 40.588, *p* < 0.01].

Exendin-4 also reduced EE over the dark cycles in male rats [F(1,15) = 24.032, *p* < 0.01] (Figure 4C). Similarly to what was observed following the GEP44 treatment, exendin-4 treatment was also associated with an increase in EE during the light cycle [F(1,15) = 40.588, *p* < 0.01].

#### EE (Females)

Consistent with our finding that GEP44 and exendin-4 both reduced core temperature in female rats, GEP44 and exendin-4 both reduced EE throughout the post-treatment period. GEP44 reduced EE over the dark cycle in female rats [F(1,15) = 26.106, *p* < 0.01] (Figure 4B). In contrast, GEP44 treatment was associated with an increase in EE during the light cycle [F(1,15) = 5.150, *p* < 0.05].

Exendin-4 also reduced EE over the dark cycle in female rats [F(1,15) = 30.036, *p* < 0.01] (Figure 4D). Similarly to what was observed following GEP44 treatment, exendin-4 treatment was also associated with an increase in EE during the light cycle [F(1,15) = 23.463, *p* < 0.01].

### 2.19. Effect of GEP44 vs. Exendin-4 on EE in Male and Female Rats

There were no differences between GEP44 and exendin-4 on EE during the light or dark cycles in male or female rats.

### 2.20. Study 2.1C: RER (Males)

We measured RER to assess the extent to which GEP44 and exendin-4 impacted lipid oxidation in male rats. GEP44 and exendin-4 both reduced RER throughout the post-treatment period. GEP44 reduced RER over both the dark cycle [F(1,15) = 99.686, *p* < 0.01] and light cycle [F(1,15) = 38.185, *p* < 0.01] in male rats (Figure 5A).

Exendin-4 also reduced RER over both the dark cycle [F(1,15) = 144.771, *p* < 0.01] and light cycle [F(1,15) = 31.887, *p* < 0.01] in male rats (Figure 5C).

#### RER (Females)

We measured RER to assess the extent to which GEP44 and exendin-4 impacted lipid oxidation in female rats. GEP44 and exendin-4 both reduced RER throughout the post-treatment period. GEP44 reduced EE over both the dark cycle [F(1,15) = 155.091, *p* < 0.01] (Figure 5B) and the light cycle [F(1,15) = 31.915, *p* < 0.01] **(**Figure 5D**)** in female rats.

Exendin-4 also reduced RER over both the dark [F(1,15) = 327.949, *p* < 0.01] and light cycles [F(1,15) = 111.788, *p* < 0.01] in female rats (Figure 5C).

### 2.21. Effect of GEP44 vs. Exendin-4 on RER in Male and Female Rats

There were no differences in response to GEP44 and exendin-4 on RER during the light and dark cycles in male and female rats.

### 2.22. Sex Differences Linked to Effects of GEP44 and Exendin-4 on Ambulatory Activity, EE, and RER

#### 2.22.1. Ambulatory Activity

##### GEP44

Two-way ANOVA demonstrated a significant overall effect of GEP44 to reduce ambulatory activity during the dark cycle [F(1,60) = 35.452, *p* < 0.01], but there was no overall effect of sex [F(1,60) = 2.497, *p* = 0.119] or any interactive effect of GEP44 and sex to reduce RER during the dark cycle [F(1,60) = 0.320, *p* = 0.350] in male and female rats. There was also a significant overall effect of GEP44 to reduce RER during the light cycle [F(1,60) = 4.893, *p* = 0.031], no overall effect of sex [F(1,60) = 1.250, *p* = 0.268], and no overall interactive effect of GEP44 and sex to reduce RER during the light cycle [F(1,60) = 0.008, *p* = 0.929] in male and female rats.

There were no significant differences in ambulatory activity (post-vehicle treatment) during the dark F(1,30) = 0.413, *p* = 0.525] or light cycles F(1,30) = 0.429, *p* = 0.518] between male and female rats. There was a tendency for GEP44 to reduce ambulatory activity more effectively in males during the dark cycle [F(1,30) = 3.051, *p* = 0.091]. However, there was no significant difference in GEP44 to impact ambulatory activity in males and females during the light cycle [F(1,30) = 1.760, *p* = 0.195].

##### Exendin-4

Two-way ANOVA demonstrated a significant overall effect of exendin-4 to reduce ambulatory activity during the dark cycle [F(1,60) = 52.564, *p* < 0.01], an overall effect of sex [F(1,60) = 4.343, *p* = 0.041] or any interactive effect of GEP44 and sex to reduce ambulatory activity during the dark cycle [F(1,60) = 0.548, *p* = 0.462] in male and female rats. There was no significant overall effect of GEP44 to reduce ambulatory activity during the light cycle [F(1,60) = 1.484, *p* = 0.228], no overall effect of sex [F(1,60) = 0.934, *p* = 0.338], and no overall interactive effect of GEP44 and sex to reduce ambulatory activity during the light cycle [F(1,60) = 0.423, *p* = 0.518] in male and female rats.

There was a near-significant difference in ambulatory activity (post-vehicle treatment) during the dark cycle F(1,30) = 3.423, *p* = 0.074] and light cycle F(1,30) = 0.699, *p* = 0.410] between male and female rats. There was no difference in response to exendin-4 on ambulatory activity in males and females during the dark cycle [F(1,30) = 0.384, *p* = 0.540]. Similarly, there was also no significant difference in response to exendin-4 on ambulatory activity in males and females during the dark cycle [F(1,30) = 1.082, *p* = 0.307] or light cycle [F(1,30) = 0.384, *p* = 0.540].

#### 2.22.2. EE

##### GEP44

Two-way ANOVA demonstrated a significant overall effect of sex [F(1,60) = 173.613, *p* < 0.01] but no overall interactive effect of GEP44 and sex to reduce EE during the dark cycle [F(1,60) = 0.497, *p* = 0.483] in male and female rats. There was also a significant overall effect of GEP44 to increase EE during the light cycle [F(1,60) = 4.204, *p* = 0.045], an overall effect of sex [F(1,60) = 135.485, *p* < 0.01] but no overall interactive effect of GEP44 and sex to increase EE during the light cycle [F(1,60) = 0.104, *p* = 0.748] in male and female rats.

Specifically, there was a significant reduction in EE (post-vehicle treatment) during both the dark [F(1,30) = 80.799, *p* < 0.01] and light cycles [F(1,30) = 74.041, *p* < 0.01] in male rats compared to female rats. In addition, GEP44 reduced EE more effectively in males during both the dark [F(1,30) = 96.293, *p* < 0.01] and light cycles [F(1,30) = 61.955, *p* < 0.01].

##### Exendin-4

Two-way ANOVA demonstrated a significant overall effect of sex [F(1,60) = 149.096, *p* < 0.01] but no overall interactive effect of exendin-4 and sex to reduce EE during the dark cycle [F(1,60) = 1.128, *p* = 0.292] in male and female rats.

There was also a significant overall effect of exendin-4 to increase EE during the light cycle [F(1,60) = 6.359, *p* = 0.014], an overall effect of sex [F(1,60) = 133.929, *p* < 0.01] but no overall interactive effect of exendin-4 and sex to increase EE during the light cycle [F(1,60) = 0.007, *p* = 0.932] in male and female rats.

Specifically, there was a significant reduction in EE (post-vehicle treatment) during both the dark [F(1,30) = 77.411, *p* < 0.01] and light cycles [F(1,30) = 59.002, *p* < 0.01] in male rats compared to female rats. Similarly, exendin-4-treated males had a reduction in EE relative to females during both the dark [F(1,30) = 72.077, *p* < 0.01] and light cycles [F(1,30) = 77.073, *p* < 0.01].

#### 2.22.3. RER

##### GEP44

Two-way ANOVA demonstrated a significant overall effect of GEP44 to reduce RER during the dark cycle [F(1,60) = 169.346, *p* < 0.01], but there was no overall effect of sex [F(1,60) = 2.437, *p* = 0.124] or any interactive effect of GEP44 and sex to reduce RER during the dark cycle [F(1,60) = 0.887, *p* = 0.350] in male and female rats.

There was also a significant overall effect of GEP44 to reduce RER during the light cycle [F(1,60) = 48.112, *p* < 0.01], no overall effect of sex [F(1,60) = 0.491, *p* = 0.486], and no overall interactive effect of GEP44 and sex to reduce RER during the light cycle [F(1,60) = 0.491, *p* = 0.486] in male and female rats.

Specifically, there was no significant difference in RER (post-vehicle treatment) during both the dark [F(1,30) = 2.552, *p* = 0.121] and light cycles [F(1,30) = 0.965, *p* = 0.334] in males compared to females. Similarly, there was also no sex difference on RER in response to GEP44 during both the dark [F(1,30) = 0.248, *p* = 0.622] and light cycles [F(1,30) = 0.000, *p* = 1.000].

##### Exendin-4

Two-way ANOVA demonstrated a significant overall effect of exendin-4 to reduce RER during the dark cycle [F(1,60) = 239.373, *p* < 0.01], no overall effect of sex [F(1,60) = 1.254, *p* = 0.267] but there was a near overall interactive effect of GEP44 and sex to reduce RER during the dark cycle [F(1,60) = 3.522, *p* = 0.065] in male and female rats.

There was also a significant overall effect of GEP44 to reduce RER during the light cycle [F(1,60) = 74.483, *p* < 0.01], no overall effect of sex [F(1,60) = 0.809, *p* = 0.809] but a near-significant overall interactive effect of GEP44 and sex to reduce RER during the light cycle [F(1,60) = 3.380, *p* = 0.071] in male and female rats.

Similarly to what we observed following GEP44 vehicle treatment, there tended to be a reduction in RER (post-exendin-4 vehicle treatment) during the dark cycle [F(1,30) = 3.450, *p* = 0.073] in male rats compared to female rats but there was no sex difference in RER (post-exendin-4 treatment) during the light cycle compared to female rats [F(1,30) = 1.908, *p* = 0.177].

### 2.23. Study 3.1: Determine the Effects of GEP44 and Exendin-4 on BAT Thermogenesis in Male and Female HFD-Fed Rats

#### 2.23.1. Study 3.1: qPCR

##### IBAT

As an additional readout of GEP44 and exendin-4-elicited thermogenic effects in IBAT, relative levels of *Ucp1*, *Adrb1*, *Adrb3*, *Gpr120*, *Ppargc1a*, *Cidea*, *Prdm16*, and *Dio2* mRNA expression were compared by PCR in response to GEP44 (50 nmol/kg) and exendin-4 (10 nmol/kg) or vehicle treatment at 2 h post-injection in male (Table 2A) and female rats (Table 2B). We recently extended these findings and found that exendin-4 resulted in a reduction in thermogenic gene expression in IBAT (consistent with the reduction in core temperature). We found that exendin-4 reduced *Adrb1* mRNA expression and increased both *Cidea* and *Ppargc1a* mRNA expression in IBAT from male rats (*p* < 0.05). In addition, we found that exendin-4 reduced *Adrb1*, *Dio2,* and *Gpr120* mRNA expression from IBAT in female rats. Similarly, we found that GEP44 reduced *Adrb1* and *Gpr120* mRNA expression from IBAT in male rats (*p* < 0.05) and it also tended to reduce *Ucp1* and *Dio2* from IBAT in male rats (0.05 < *p* < 0.1). In addition, it also reduced *Adrb1*, *Ucp1, Dio2,* and *Gpr120* mRNA expression in female rats (*p* < 0.05) and tended to increase *Prdm16* mRNA expression from IBAT of female rats (0.05 < *p* < 0.1).

##### IWAT

In addition, we found that both exendin-4 and GEP44 stimulated *Gpr120* mRNA expression in IWAT from male rats (*p* < 0.05; Table 2C). Furthermore, GEP44 also stimulated *Ppargc1a* mRNA expression in IWAT of male rats (*p* < 0.05), while exendin-4 tended to stimulate *Ppargc1a* mRNA expression from IWAT in male rats (0.05 < *p* < 0.1). In female rats, exendin-4 tended to both reduce *Dio2* and increase *Gpr120* mRNA expression in IWAT (Table 2D). In addition, we found that GEP44 reduced *Dio2* mRNA expression from IWAT.

While the qPCR data were normalized to the housekeeping gene, *Nono*, similar results were obtained when normalizing the data to *Gapdh*.

Together, these findings suggest that GLP-1 receptor agonists result in a reduction in BAT thermogenesis in both male and female rats. Given that we also found this to be the case with EE, these findings suggest that both exendin-4- and GEP44-elicited weight loss is primarily due to reductions in energy intake. Furthermore, the reductions in BAT thermogenesis and EE associated with GEP44 and exendin-4 might be secondary to a reduction in diet-induced thermogenesis or might indicate an important mechanism to limit the overall efficacy of GEP44 and exendin-4 to prevent further weight loss.

### 2.24. Study 3.2A-B: (A) Blood Glucose and (B) Plasma Hormones

Consistent with previous findings in rats, we found that GEP44 tended to reduce tail vein glucose (collected at 2-h post-injection) in male rats (Table 3A; *p* = 0.099). In female rats, GEP44 produced a significant reduction in tail vein glucose [F(1,13) = 8.450, *p* = 0.012] (Table 3B). In addition, exendin-4 reduced fasting tail vein glucose in male rats [F(1,10) = 5.365, *p* = 0.043] (Table 3A), but exendin-4 was not effective at reducing fasting tail vein glucose in female rats (Table 3B; *p* = NS). The reduction in blood glucose in response to GEP44 in male and female rats did not appear to be mediated by glucagon, as there was no significant change in glucagon in response to GEP44. However, glucagon was reduced in response to exendin-4 in male rats [F(1,10) = 5.284, *p* = 0.044] but not in female rats, thus raising the possibility that the reduction in blood glucose in response to exendin-4 in male rats may be mediated, in part, by the reduction in glucagon in male rats. Neither GEP44 nor exendin-4 reduced plasma leptin in male or female rats (Table 3A,B; *p* = NS). GEP44 increased plasma insulin in both male [F(1,11) = 5.270, *p* = 0.042] and female rats [F(1,13) = 9.232, *p* = 0.010], consistent with what we had observed in male DIO mice [19]. Exendin-4 also increased plasma insulin in female [F(1,12) = 11.704, *p* = 0.005] but not male rats. In addition, GEP44 also reduced plasma adiponectin in male [F(1,11) = 32.059, *p* < 0.01] and female rats [F(1,13) = 33.492, *p* < 0.01]. Similarly, exendin-4 also reduced plasma adiponectin in both male [F(1,10) = 14.979, *p* = 0.003] and female rats [F(1,12) = 13.017, *p* = 0.004]. There was a tendency for GEP44 to increase irisin in male rats (*p* = 0.064) and female rats (*p* = 0.065). Likewise, exendin-4 also tended to increase irisin in male rats (*p* = 0.088) and female rats (*p* = 0.057). GEP44 also increased free fatty acids in female [F(1,13) = 11.534, *p* = 0.005] but not in male rats. Similarly, exendin-4 tended to increase FFA in female (*p* = 0.057) but not in male rats. GEP44 also elevated total cholesterol in male rats [F(1,11) = 5.483, *p* = 0.039] and tended to elevate total cholesterol in female rats (*p* = 0.080). In contrast, exendin-4 failed to significantly alter total cholesterol levels in either male or female rats. Lastly, both GEP44 [F(1,13) = 18.195, *p* = 0.001] and exendin-4 [F(1,12) = 30.669, *p* < 0.01] produced a robust increase in FGF-21 in female rats, similar to what has been reported in other studies in male mice [20].

## 3. Discussion

The goal of the current study was to determine whether the novel chimeric triple agonist, GEP44, which targets GLP-1R, Y1R, and Y2R, reduced EE and RER at doses that elicit reductions in core temperature, energy intake, and weight loss in adult male and female HFD-fed rats. As observed in the mouse model [19], GEP44 also reduced core temperature, energy intake, and body weight in both male and female HFD-fed rats. In contrast to our initial hypothesis, we found that GEP44 reduced EE and RER at doses that reduced core temperature, body weight, and energy intake. In addition, we found there to be sex differences with respect to the effects of GEP44 and exendin-4 on core temperature and body weight, with GEP44 being able to produce more profound effects in male rats. Similarly to what we have previously reported in male and female rats [14,21,22] and mice [19], GEP44-elicited reductions in body weight appear to be driven largely by reductions in energy intake, but, here, we also found that it increased lipid oxidation (denoted by reduced RER). Furthermore, our finding that GEP44 also reduced EE might be secondary to a reduction in diet-induced thermogenesis or might indicate an important mechanism to limit the overall efficacy of GEP44 to prevent further weight loss.

Most studies that have administered GLP-1R agonists systemically have either found a decrease or no change in EE or have failed to measure both core temperature and EE in the same group of animals. We extend previous findings from our laboratory [14,21] to demonstrate that GEP44 reduces core temperature (a surrogate measure of EE) in both male and female rats, similar to what we described earlier in response to exendin-4 treatment. Our findings in male and female rats are also largely consistent with our recently published study in male and female DIO mice [19]. Previous studies have also reported that peripheral administration of the GLP-1R agonist, exendin-4, reduced core temperature for up to 4 h in rats [23]. In addition, others have reported that peripheral administration of exendin-4 decreased either resting EE over 2 and 21 h in adult male wild-type mice [17] or EE at 4 h post-injection in rats [24]. These findings have translated to humans, as others have also found that liraglutide decreased resting EE after 4 weeks of treatment in humans with type 2 diabetes [25]. On the other hand, others found that semaglutide decreased dark phase EE through six days of treatment in DIO mice, but these findings were not significant after adjusting for lean mass [26]. Similarly, others also found that semaglutide reduced resting EE after 12 weeks of treatment in humans but these effects were not significant after adjusting for lean mass [27].

Similar to what others have reported on core temperature being lower in response to thermoneutral temperatures or a cold challenge in male rats, we found that core temperature was lower in male rats following vehicle administration or during baseline (untreated) measurements. McDonald and colleagues [28] determined that female rats had elevated BAT mass and more consistent guanosine 5′-diphosphate (GDP) binding to mitochondria derived from BAT during the aging process, while male rats had both reduced BAT mass and GDP binding. Liu and colleagues recently reported that the ability of central oxytocin to reduce food intake in female rats was blocked during proestrus [29]. Furthermore, Liu revealed in subsequent studies that estrogen replacement blocked the effectiveness of central OT in reducing food intake in ovariectomized rats [29]. Collectively, these sex differences with respect to core temperature and differences in the estrous cycle might explain, in part, why we found what appeared to be an enhanced effectiveness of GEP44 and exendin-4 to reduce core temperature and potentially elicit weight loss in male rats relative to female rats. Lopez-Ferreras and colleagues have found there to be a more attenuated reduction in sucrose reward and active lever presses to obtain the reward in response to the administration of exendin-4 into the lateral hypothalamus [30]. A recent review by Borhers and Skibicka [31] indicated that differential distribution of GLP-1R between males and females might explain potential sex differences in the response to GLP-1R analogs in rodent models (for review, see [31]).

Consistent with our findings that GEP44 and exendin-4 reduced core temperature and EE, we found that both GEP44 and exendin-4 reduced thermogenic markers (biochemical readout of BAT thermogenesis) within IBAT and T_IBAT_ (functional readout of BAT thermogenesis). These findings are consistent with the notion that BAT thermogenesis is important in maintaining core temperature in other contexts such as fever or stress [32,33]. Our findings, however, that exendin-4 and GEP44 reduced thermogenic markers within IBAT are consistent with what others have reported following intraperitoneal administration of exendin-4 in a rat model [24]. Krieger found that exendin-4 reduced BAT *Adrb3* mRNA expression in rats. While we did not find a change in *Adrb3* mRNA expression in our rat model, we did find that exendin-4 reduced IBAT *Adrb1* mRNA expression in IBAT in both male and female DIO rats. In addition, exendin-4 reduced IBAT *Adrb1*, Ucp1, *Dio2,* and *Gpr120* mRNA expression in female rats. In addition, GEP44 also reduced IBAT Adrb1 mRNA expression in female rats, but, similar to what we found following exendin-4 in female rats, GEP44 also reduced IBAT *Adrb1*, *Ucp1*, *Dio2*, and *Gpr120* mRNA expression. In addition, we found that GEP44 also stimulated IWAT *Gpr120* and *Ppargc1a* mRNA expression in male rats while it, like exendin-4, reduced *Dio2* mRNA expression in female rats. Overall, our findings suggest that systemic administration of both GEP44 and exendin-4 may reduce EE and BAT and WAT thermogenesis through both overlapping and distinct mechanisms in male and female rats. It remains to be determined whether the browning or beiging of WAT is playing a major role in how GEP44 reduces body weight in male DIO rats.

The findings from the current study demonstrating that GEP44 and exendin-4 reduced both core temperature and ambulatory activity in both male and female rats are consistent with what we recently reported in both male and female DIO mice [19]. Similarly, others have also found that systemic administration of exendin-4 reduces locomotor activity in lean male Sprague Dawley rats [34,35]. While others did not find any change in activity in response to systemic exendin-4 treatment in lean male Sprague Dawley rats [23], it might be possible that differences in timing of injections, body weight, animal model, and/or dosing may explain, in part, the different results between studies. Given that we found reductions in both core temperature and gross motor activity in male rats, our data raise the possibility that reductions in shivering and non-shivering thermogenesis in skeletal muscle [16] and/or spontaneous physical activity-induced thermogenesis [15] may have contributed to the hypothermic effects of both GEP44 and exendin-4 in male rats. Collectively, our findings indicate that GEP44 and exendin-4 may reduce 1) activity in male rats but not in female rats and 2) core temperature in both male and female rats.

Similarly to what we reported in DIO mice [19], the findings from the current study showed that both GEP44 and exendin-4 treatment improved glucose homeostasis in the rat model. GEP44 increased fasting plasma insulin in both male and female HFD rats. This effect was also observed following exendin-4 treatment in female HFD rats and is consistent with the reduction in blood glucose (tail vein) in response to GEP44 in male and female rats, as well as the reduction in blood glucose in response to exendin-4 in male rats. It is possible that, similar to what we found in the mouse model [19], reductions in glucagon may contribute, in part, to the glucose-lowering effects. Increased FGF-21 levels (not measured in male rats) in response to both GEP44 and exendin-4 in female rats may also contribute, in part, to the reduction in blood glucose [36]. These findings are similar to what has been reported in other studies in male mice [20]. We also found evidence that both GEP44 and exendin-4 may also increase lipolysis, as indicated by the increase in plasma FFA in female HFD-fed rats. Similarly to our findings, Nanchani reported that exendin-4 treatment appeared to reduce adiponectin levels in male rats, although the results were not significant [37]. How adiponectin might contribute to the metabolic or glucoregulatory effects of both GEP44 and exendin-4 remains to be determined. In contrast to what we observed in male and female mice [19], GEP44 and exendin-4 appeared to reduce total cholesterol levels. Elfers and colleagues recently reported that cholesterol levels in response to GEP44 were elevated relative to male DIO rats that were pair-fed to the GEP44 treatment group [22], yet there did not appear to be any differences between vehicle and the GEP44 treatment group. Note that cholesterol levels appeared to increase in response to GEP44 treatment relative to vehicle but it was not significant. It remains to be determined whether the differences in our studies and the earlier studies represent differences in species or paradigm differences.

One limitation of our studies is the lack of weight-restricted or pair-fed controls for studies that assessed the effects of GEP44 and exendin-4 on IBAT and IWAT gene expression and EE in adult DIO male and female rats. As a result, it is possible that the effects of GEP44 and exendin-4 on the reduction in thermogenic gene expression or EE in HFD-fed rats may also be due, in part, to the reduction in diet-induced thermogenesis. Future studies incorporating the use of similar measurements in fasted animals will be helpful in more fully establishing the role of GEP44 and exendin-4 on thermogenesis and EE in male and female rats.

## 4. Materials and Methods

### 4.1. Animals

Adult male (~600–907 g; *n*= 19 at study onset) and female (~275–479 g; *n* = 23 at study onset) Long–Evans rats were initially obtained from Charles River Laboratories (Wilmington, MA, USA) and maintained for at least 4 months on a high-fat diet (HFD) prior to study onset. All animals were housed individually in Plexiglas cages in a temperature-controlled room (22 ± 2 °C) under a 12:12 h light–dark cycle. All rats were maintained on a 1 a.m. (lights on)/1 p.m. (lights off) light cycle. Rats had ad libitum access to water and a HFD providing 60% kcal from fat (Research Diets, D12492, New Brunswick, NJ, USA). Animals were maintained on a 1-h fast during the week prior and throughout the vehicle and drug treatment phases of Studies 1.1–2.1 as listed below. The research protocols were approved by the Institutional Animal Care and Use Committee of the Veterans Affairs Puget Sound Health Care System (VAPSHCS) in accordance with NIH’s *Guide for the Care and Use of Laboratory Animals* (NAS, 2011) [38].

### 4.2. Drug Preparation

GEP44 was synthesized in the Doyle lab as previously described [14]. Fresh solutions of GEP44 and exendin-4 (ENZO; Farmingdale, NY, USA) were prepared, frozen, and thawed prior to the onset of each experiment (Studies 1–3).

### 4.3. Implantation of Implantable Telemetry Devices (PDT-4000 E-Mitter) into the Abdominal Cavity

Animals were anesthetized with isoflurane and subsequently underwent sham surgery (no implantation) or received implantations of a sterile PDT-4000 E-Mitter (23 mm long × 8 mm wide; Starr Life Sciences Company, Oakmont, PA, USA) into the intraperitoneal cavity. The abdominal opening was closed with 4-0 Vicryl; Ethicon, Bridgewater Township, NJ, USA) absorbable suture and the skin was closed with 4-0 monofilament nonabsorbable suture. Vetbond glue (3M; St. Paul, MN, USA)was used to seal the wound and bind any tissue together between the sutures. Sutures were removed within two weeks after the PDT-4000 E-Mitter implantation. All PDT-4000 E-Mitters were confirmed to have remained within the abdominal cavity at the conclusion of the study.

### 4.4. Acute SC Injections of GEP44 and Exendin-4

GEP44 (or saline vehicle; 1 mL/kg injection volume) or exendin-4 were administered immediately prior to the start of the dark cycle following 1 h of food deprivation. Rats underwent all treatments over the course of a 2-week period. The study design consisted of one round of a 2-day baseline phase (vehicle-treated), a 2-day treatment phase (single dose repeated over 2 days), and a washout phase (approximately 3-day washout period for Study 1 and Study 2). During the drug treatment phase, one half of the animals received exendin-4 (10 nmol/kg) while the other half received GEP44 (50 nmol/kg). Treatments were counterbalanced so that each animal received exendin-4 (10 nmol/kg/day) during the first week and received GEP44 (50 nmol/kg) during the second week and vice versa. Body weight and energy intake were assessed daily at approximately 1 h prior to the start of the dark cycle.

### 4.5. Body Composition

Determinations of lean body mass and fat mass were made on conscious rats by quantitative magnetic resonance using an EchoMRI 4-in-1-700^TM^ instrument (Echo Medical Systems, Houston, TX, USA) at the VAPSHCS Rodent Metabolic Phenotyping Core.

## 5. Study Protocols

### 5.1. Study 1.1: Determine the Effects of GEP44 and Exendin-4 on Signs of Pain or Discomfort, Body Weight, and Energy Intake in Male and Female HFD-Fed Rats

#### 5.1.1. Signs of Pain or Discomfort

Adult male (*n* = 19/group) and female rats (*n* = 23/group) were observed for facial signs of pain or discomfort (narrow eyes and flattened or elongated nose) during the period immediately following GEP44 and exendin-4 treatment in male and female HFD-fed rats.

#### 5.1.2. Changes in Body Weight and Energy Intake

Prior to the onset of vehicle and drug treatment within each sex, both treatment groups were matched for body weight and adiposity. The study design is shown in Appendix A and was described earlier under “Acute SC injections of GEP44 and Exendin-4”. By design, male rats were DIO as determined by both body weight (730.2 ±19.1 g) and adiposity (219 ± 14 g fat mass; 29.6 ± 1.2% adiposity) after maintenance on the HFD for approximately 4 months. As expected, female HFD-fed rats were overweight/obese as confirmed by both body weight (340.8 ± 9.9 g) and adiposity (79.6 ± 8.2 g fat mass; 22.5 ± 1.6% adiposity). The selective GLP-1R agonist, exendin-4, was included as a control to assess GLP-1R-mediated effects on energy intake [17]. Body weight and energy intake were measured daily. Body weight loss was calculated from the total change in body weight over a 2-day drug treatment period relative to body weight prior to drug treatment on day 1. Energy intake data reflects the cumulative intake throughout the 2-day vehicle and 2-day drug treatment period. Total cumulative food intake was measured daily from each home cage hopper. The difference in weight of high-fat diet pellets from each hopper before and 48-h post-treatment intervention represented the amount of food ingested over the 2-day treatment period. Food intake (grams) was subsequently converted to units of energy intake (5.24 kcal/g).

### 5.2. Study 1.2A-B: Determine the Effects of GEP44 and Exendin-4 on (A) Core Temperature and (B) Gross Motor Activity in Male and Female HFD-Fed Rats Using an E-Mitter Telemetry System

#### Study 1.2A-B: Changes in (A) Core Temperature and (B) Gross Motor Activity

Adult male (*n* = 15/group) and female rats (*n* = 7/group) were used for these studies. The protocol for measuring core temperature and gross motor activity has been described previously [39]. Briefly, core temperature (a surrogate marker of EE) and gross motor activity were recorded from each rat in the home cage every 5 min throughout the study. The last hour of the light cycle (during which time drug administration and assessments of energy intake and body weight occurred) was excluded from the telemetry analysis. Core temperature and activity were averaged throughout each sequential 2-day vehicle and 2-day drug treatment period.

### 5.3. Study 2.1A-C: Determine the Effects of GEP44 and Exendin-4 on (A) Ambulatory Activity, (B) EE, and (C) RER and in Male and Female HFD-Fed Rats Using the CLAMS-HC Indirect Calorimetry System

#### Study 2.1A-C: Changes in (A) Ambulatory Activity, (B) EE, and (C) RER

A subset of the same male (*n* = 16/group) and female rats (*n* = 16/group) from Studies 1.1 and 1.2 were used for these studies. The protocol for measuring EE has been previously described [40]. Similarly to Study 1.1, male and female HFD-fed rats underwent a 2-day vehicle period and 2-day drug treatment (GEP44 at 50 nmol/kg vs. the selective GLP-1R agonist, exendin-4 at 10 nmol/kg). Following an approximate 3-day washout period, treatments were given in a counterbalanced design so that each animal received both treatments over the 2-week experimental period. Rats were acclimated to indirect calorimetry cages prior to the study and data collection. EE measures were obtained using a computer-controlled indirect calorimetry system (CLAMS-HC, Columbus Instruments, Columbus, OH, USA) located in the Rodent Metabolic Phenotyping Core at the VAPSHCS. Animals were individually housed throughout the testing period in a home-cage-like setting (Tecniplast cage 1250 (Tecniplast; West Chester, PA, USA); floor area 580 cm^2^), in a light-and temperature-controlled enclosure (Powers Scientific, Pipersville, PA, USA), with the temperature set to 27 ± 1 °C, and a 12 h light/dark schedule. Irradiated corncob bedding and environmental enrichment (e.g., nylabone, gnawing sticks) were supplied on the floor of the cage. Feeders and water bottles were suspended from load cells, which enabled continuous measurement of food and water intake. Physical activity, cage temperature, and humidity were all continuously measured, as was respiratory rate. Respiratory gases were measured by an integrated zirconia oxygen sensor and single-beam non-dispersed infrared CO_2_ sensor. Gas sensors were calibrated at the beginning of each experiment with a span gas containing known concentrations of O_2_ and CO_2_ with a balance of N2 (PraxAir, Tacoma, WA, USA). O_2_ consumption (VO_2_) and CO_2_ (VCO_2_) production were measured for each rat for 20 s with 9 measurements per animal per hour. Incurrent air reference values were determined after measuring every sixteen cages. RQ was calculated as the ratio of CO_2_ production over O_2_ consumption. EE was calculated by Columbus Instrument’s software using the equation VO_2_ × (3.815 × 1.232 RQ) and was expressed in units of kilocalories per hour. [Note: the Columbus system does not use the Weir equation, namely, kcal/h = 60 × (0.003941 × VO_2_ + 0.001106 × VCO_2_.)] Total and ambulatory activity was determined simultaneously with the collection of the calorimetry data. Consecutive adjacent infrared beam breaks in the X and Y axes, i.e., the length and width of the cage, were scored as activity counts, and the tally was recorded every 6.7 min. Data acquisition, instrument control, and data processing were coordinated by Oxymax software (v5.68, Columbus Instruments).

### 5.4. Study 3.1 Determine the Effects of GEP44 and Exendin-4 on Brown Adipose Tissue (BAT) Thermogenesis in Male and Female HFD-Fed Rats

#### 5.4.1. Study 3.1: Tissue Collection for Quantitative Real-Time PCR (qPCR)

The same male (*n* = 19 total; *n* = 6–7/group) and female rats (*n* = 7–8/group) from Studies 1.1–2.1 were used for these studies. Following a washout period of approximately 16 days at the end of Study 2.1, animals were fasted for 4 h prior to being treated once daily for 3 consecutive days with either vehicle, GEP44 or exendin-4. The more extended washout period (16 days) coincided with the process of receiving animal care protocol approval for additional treatments with GEP44 and exendin-4. Tissue [interscapular brown adipose tissue (IBAT)] was collected from 6 h fasted rats at 2 h post-injection. Rats were euthanized with an overdose of ketamine cocktail at 2 h post-injection. The tissue was rapidly removed, wrapped in foil, and frozen in liquid N2. Samples were stored frozen at −80 °C until analysis. IBAT was collected within a 5 h window towards the end of the light cycle (9:00 a.m.–2:00 p.m.) as previously described in DIO CD^®^IGS/Long–Evans rats and C57BL/6J mice [41,42,43].

#### 5.4.2. Study 3.1: qPCR

RNA extracted from samples of IBAT was analyzed using the RNeasy Lipid Mini Kit (Qiagen Sciences Inc, Germantown, MD), followed by reverse transcription into cDNA using a high-capacity cDNA archive kit (Applied Biosystems, Foster City, CA, USA). Quantitative analysis for relative levels of mRNA in the RNA extracts was measured in duplicate by qPCR on an Applied Biosystems 7500 Real-Time PCR system (Thermo Fisher Scientific, Waltham, MA) using the following TaqMan^®^ probes (Thermo Fisher Scientific Gene Expression Assay probes; Table 4: rat *Nono,* glyceraldehyde-3-phosphate dehydrogenase (*Gapdh*), uncoupling protein-1 (*Ucp1*), β1-adrenergic receptor (*Adrb1*), β3-adrenergic receptor (*Adrb3*), type 2 deiodinase (*Dio2*), PR domain containing 16 (*Prdm16*), G-protein coupled receptor 120 (*Gpr120*), cell death-inducing DNA fragmentation factor alpha-like effector A (*Cidea*), and peroxisome proliferator-activated receptor gamma coactivator 1 alpha (*Ppargc1a)*. Relative amounts of target mRNA were determined using the Comparative C_T_ or 2^−ΔΔCT^ method [44] following adjustment for the housekeeping gene, *Nono* and *Gapdh*. Specific mRNA levels of all genes of interest were normalized to the cycle threshold value of Nono and *Gapdh* mRNA in each sample and expressed as changes normalized to controls (vehicle treatment).

### 5.5. Study 3.2A-B: Determine the Effects of GEP44 and Exendin-4 on Blood Glucose and Plasma Hormones

#### 5.5.1. Study 3.2: Blood Collection

Following a 6-h fast, blood samples [up to 2 mL] from animals in Study 3.1 were collected by a cardiac stick in chilled K2 EDTA Microtainer Tubes (Becton-Dickinson, Franklin Lakes, NJ, USA) at 2-h post-injection. Whole blood was centrifuged at 6000 rpm for 1.5 min at 4 °C; plasma was removed, aliquoted, and stored at −80 °C for subsequent analysis.

#### 5.5.2. Study 3.2A: Blood Glucose Measurements

Following at least a 16-day washout period, animals were divided into three groups (VEH, GEP44, and exendin-4) matched for body weight and adiposity. Animals were fasted for 4 h prior to receiving vehicle, GEP44 or exendin-4 once daily for 3 consecutive days. On day 3, both tail vein glucose and tissues were collected at 2-h post-injection. Blood was collected from 6 h fasted animals at 2-h post-injection in a subset of rats for glucose measurements by tail vein nick and measured using a glucometer (AlphaTRAK 2, Abbott Laboratories, Abbott Park, IL, USA) [41,42].

#### 5.5.3. Study 3.2B: Plasma Hormone Measurements

Plasma leptin and insulin were measured using electrochemiluminescence detection [Meso Scale Discovery (MSD^®^), Rockville, MD, USA] using established procedures [42,45]. The intra-assay coefficient of variation (CV) for leptin was 2.8%. Intra-assay CV for insulin was 2.7%. Plasma glucagon (Mercodia, Winston Salem, NC, USA), fibroblast growth factor-21 (FGF-21) (R&D Systems, Minneapolis, MN, USA), and irisin (AdipoGen, San Diego, CA, USA) levels were determined by ELISA. The intra-assay CV for glucagon was 1.6%. The intra-assay CV for FGF-21 was 2.7%. The intra-assay CV for irisin was 6.9% for mice (not obtained for rats). Plasma adiponectin was also measured using ELISA (Alpco, Salem, NH, USA) using established procedures [42,45]. The intra-assay CV for adiponectin was 1.7%. The range of detectability for each hormone is listed in Appendix A. The data were normalized to historical values using a pooled plasma quality control sample that was assayed in each plate.

## 6. Statistical Analyses

All results are expressed as mean ± SE. Planned comparisons within respective sex between vehicle and drug (plasma measures, tail vein glucose, and gene expression data) involving between-subjects design were made using one-way ANOVA followed by a post hoc Fisher’s least significant differences test. Planned comparisons within respective sex to examine treatment means of vehicle and drug (body weight loss, energy intake, core temperature, activity, and EE) involving within-subjects designs were made using a one-way repeated-measures ANOVA followed by a post hoc Fisher’s least significant differences test. In addition, comparisons between multiple groups to examine drug or sex differences in body weight loss and energy intake were made using two-way ANOVA followed by a post hoc Fisher’s least significant difference test. Analyses were performed using the statistical program SYSTAT (Version 13; Systat Software, Point Richmond, CA, USA). Differences were considered significant at *p* < 0.05, 2-tailed.

## 7. Conclusions

In summary, the results presented in this manuscript highlight the effects of the novel chimeric peptide, GEP44, on EE, RER, core temperature, ambulatory and gross motor activity, body weight, and energy intake in adult male DIO and female HFD-fed rats at doses that have not been found to elicit visceral illness in rats (as determined by facial observations of nausea or pain [14] and kaolin intake [21]) or emesis in shrews [14,21]. In contrast to our initial hypothesis that the strong reduction in body weight in response to GEP44 is partially related to the stimulation of EE, the findings from the current study showed that GEP44 reduced EE and RER at doses that reduced core temperature, body weight, and energy intake. The anti-obesity effects of GEP44 and exendin-4 appear to be mediated, in part, by increased lipid oxidation and reductions in energy intake. The results from the current study further demonstrate that there may be overlapping and distinct mechanisms that mediate the effects of GEP44 and exendin-4 on BAT thermogenesis and EE in male and female rats. Furthermore, our finding that GEP44 reduced EE might be secondary to a reduction in diet-induced thermogenesis or might be an important safety override mechanism to prevent clinically extreme weight loss. The findings to date indicate that GEP44 is a promising drug that overcomes the adverse side effects associated with existing GLP-1R agonist medications [5,46] to treat obesity and/or T2DM.

## Figures and Tables

**Figure 1 ijms-26-03032-f001:**
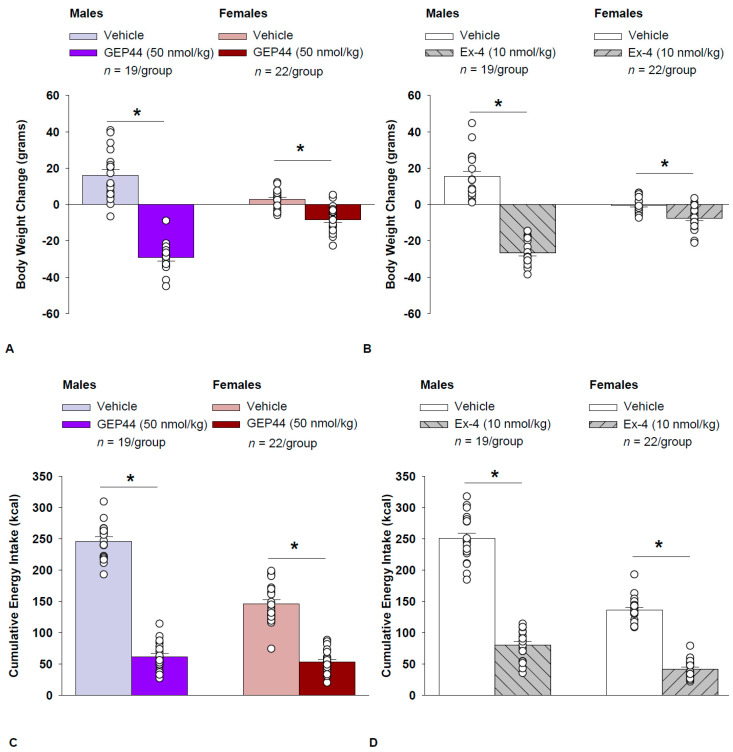
(**A**–**D**) Effects of the chimeric peptide, GEP44, or selective GLP-1R agonist, exendin-4, on body weight and energy intake in male and female HFD-fed rats. Male (*n* = 19/group) and female (*n* = 22/group) rats were maintained on HFD (60% kcal from fat) for at least 4 months prior to being implanted with PDT-4000 telemetry devices into the abdominal cavity. During this study, animals remained in their home cages and were maintained on a 1-h fast. Animals received SC injections of vehicle (sterile saline/water) followed by GEP44 (50 nmol/kg; 1 mL/kg injection volume) or exendin-4 within 15 min prior to the start of the dark cycle in a counterbalanced design. (**A**) Effect of GEP44 on change in body weight in male and female HFD-fed rats; (**B**) Effect of exendin-4 on change in body weight in male and female HFD-fed rats; (**C**) Effect of GEP44 on energy intake in male and female HFD-fed rats; (**D**) Effect of exendin-4 on energy intake in male and female HFD-fed rats. The change in body weight and cumulative energy intake reflects the total change in both measurements over the sequential 2-day vehicle and 2-day drug treatment period. Data are expressed as mean ± SEM. * *p* < 0.05 GEP44 or exendin-4 vs. vehicle.

**Figure 2 ijms-26-03032-f002:**
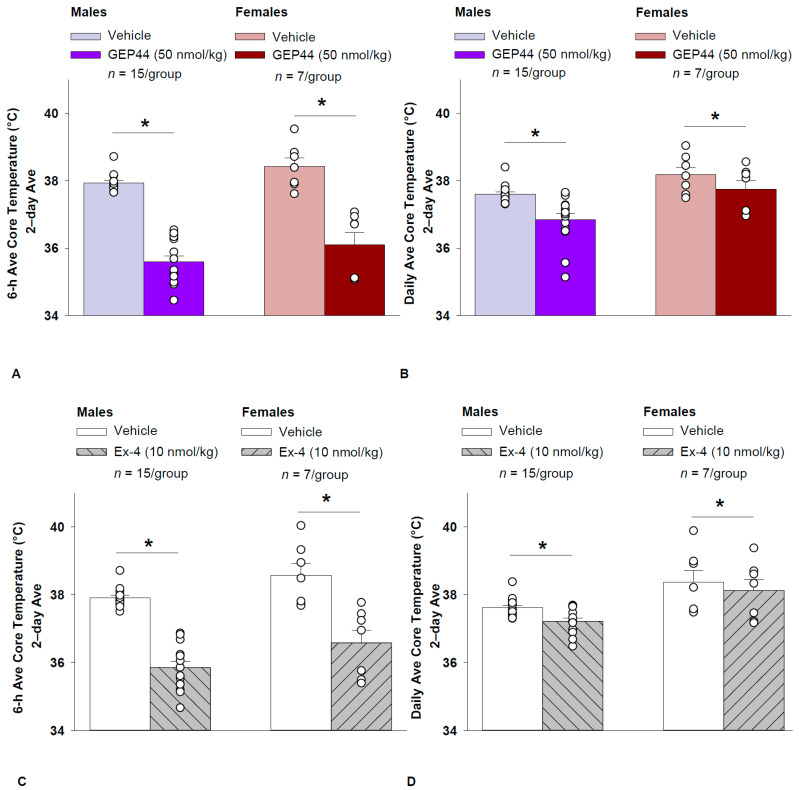
(**A**–**D**) Effects of the chimeric peptide, GEP44, or selective GLP-1R agonist, exendin-4, on core temperature in male and female HFD-fed rats. Male (*n* = 15/group) and female (*n* = 7/group) rats were maintained on HFD (60% kcal from fat) for at least 4 months prior to being implanted with PDT-4000 telemetry devices into the abdominal cavity. During this study, animals remained in their home cages and were maintained on a 1-h fast. Animals received SC injections of vehicle (sterile saline/water) followed by GEP44 (50 nmol/kg; 1 mL/kg injection volume) or exendin-4 within 15 min prior to the start of the dark cycle in a counterbalanced design. (**A**) Effect of GEP44 on 6-h core temperature in male and female HFD-fed rats. (**B**) Effect of GEP44 on daily (23 h) core temperature in male and female HFD-fed rats. (**C**) Effect of exendin-4 on 6-h core temperature in male and female HFD-fed rats. (**D**) Effect of exendin-4 on daily (23 h) core temperature in male and female HFD-fed rats. Core temperature was averaged at 6-h and 23-h post-treatment periods over the 2-day vehicle and 2-day drug treatment period. Data are expressed as mean ± SEM. * *p* < 0.05 GEP44 or exendin-4 vs. vehicle.

**Figure 3 ijms-26-03032-f003:**
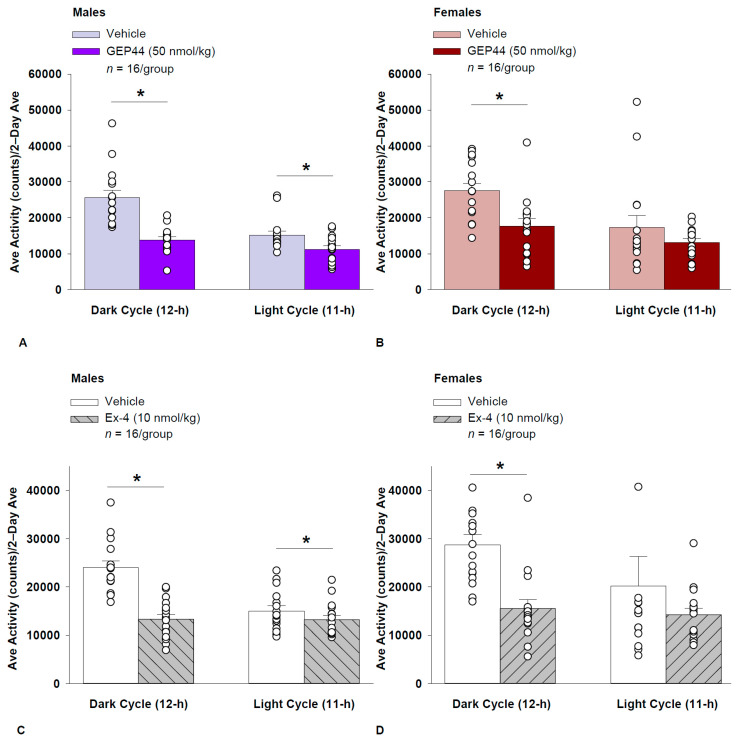
(**A**–**D**) Effects of the chimeric peptide, GEP44, or selective GLP-1R agonist, exendin-4, on ambulatory activity in male and female HFD-fed rats. Male (*n* = 16/group) and female (*n* = 16/group) rats were maintained on HFD (60% kcal from fat) for at least 4 months prior to being implanted with PDT-4000 telemetry devices into the abdominal cavity. During this study, animals remained in their home cages and were maintained on a 1-h fast. Animals received SC injections of vehicle (sterile saline/water) followed by GEP44 (50 nmol/kg; 1 mL/kg injection volume) or exendin-4 within 15 min prior to the start of the dark cycle in a counterbalanced design. (**A**/**B**) Effect of GEP44 on ambulatory activity during the light and dark cycle periods in (**A**) male and (**B**) female HFD-fed rats. (**C**/**D**) Effect of exendin-4 on ambulatory activity during the light and dark cycle periods in (**C**) male and (**D**) female HFD-fed rats. Ambulatory activity was averaged over the light (11 h) and dark (12 h) cycles across the 2-day vehicle and 2-day drug treatment periods. Data are expressed as mean ± SEM. * *p* < 0.05 GEP44 or exendin-4 vs. vehicle.

**Figure 4 ijms-26-03032-f004:**
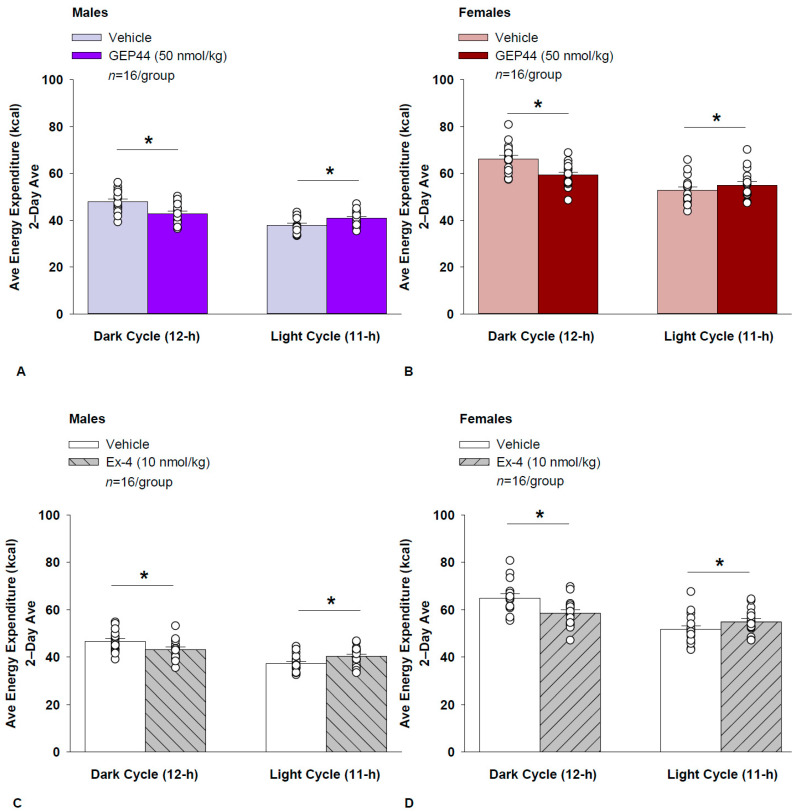
(**A**–**D**) Effects of the chimeric peptide, GEP44, or selective GLP-1R agonist, exendin-4, on EE (indirect calorimetry) in male and female HFD-fed rats. Male (*n* = 16/group) and female (*n* = 16/group) rats were maintained on HFD (60% kcal from fat) for at least 4 months prior to being implanted with PDT-4000 telemetry devices into the abdominal cavity. Animals were transferred to individual metabolic cages that were part of the CLAMS-HC indirect calorimetry system prior to receiving SC injections of vehicle (sterile saline/water) followed by GEP44 (50 nmol/kg; 1 mL/kg injection volume) or exendin-4 within 1.5 h prior to the start of the dark cycle in a counterbalanced design. (**A**/**B**) Effect of GEP44 on EE during the light and dark cycle periods in (**A**) male and (**B**) female HFD-fed rats. (**C**/**D**) Effect of exendin-4 on EE during the light and dark cycle periods in (**C**) male and (**D**) female HFD-fed rats. EE was averaged over the light (11 h) and dark (12 h) cycles across the 2-day vehicle and 2-day drug treatment periods. Data are expressed as mean ± SEM. * *p* < 0.05 GEP44 or exendin-4 vs. vehicle.

**Figure 5 ijms-26-03032-f005:**
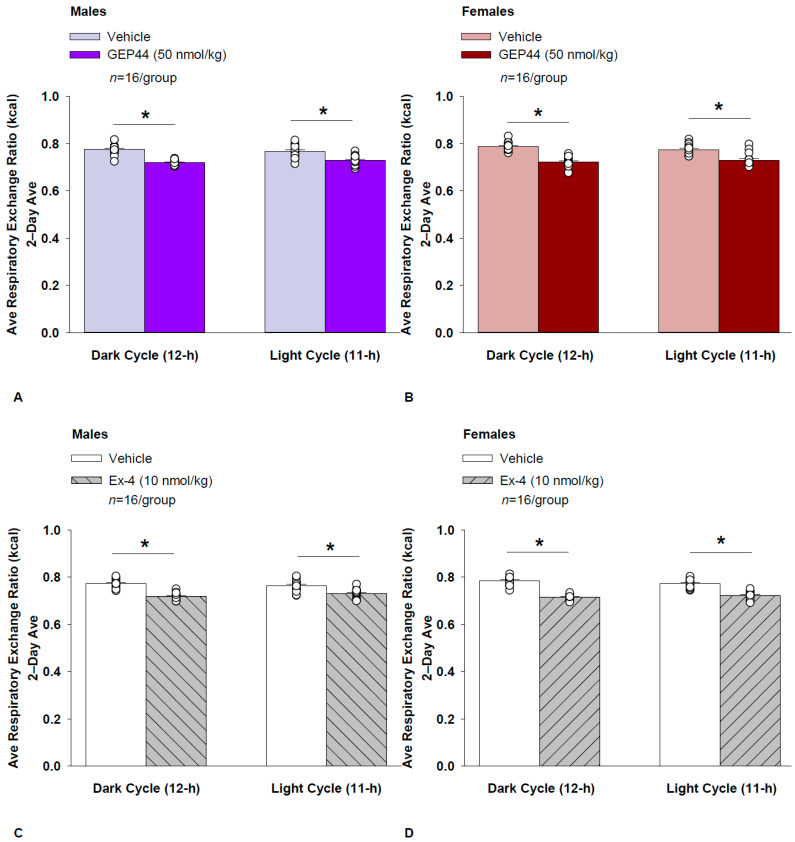
(**A**–**D**) Effects of the chimeric peptide, GEP44, or selective GLP-1R agonist, exendin-4, on RER (indirect calorimetry) in male and female HFD-fed rats. Male (*n* = 16/group) and female (*n* = 16/group) rats were maintained on HFD (60% kcal from fat) for at least 4 months prior to being implanted with PDT-4000 telemetry devices into the abdominal cavity. Animals were transferred to individual metabolic cages that were part of the CLAMS-HC indirect calorimetry system prior to receiving SC injections of vehicle (sterile saline/water) followed by GEP44 (50 nmol/kg; 1 mL/kg injection volume) or exendin-4 within 1.5 h prior to the start of the dark cycle in a counterbalanced design. (**A**/**B**) Effect of GEP44 on RER during the light and dark cycle periods in (**A**) male and (**B**) female HFD-fed rats. (**C**/**D**) Effect of exendin-4 on RER during the light and dark cycle periods in (**C**) male and (**D**) female HFD-fed rats. RER was averaged over the light (11 h) and dark (12 h) cycles across the 2-day vehicle and 2-day drug treatment periods. Data are expressed as mean ± SEM. * *p* < 0.05 GEP44 or exendin-4 vs. vehicle.

**Table 1 ijms-26-03032-t001:** Observational signs of pain or discomfort following GEP44 and exendin-4 treatment in male and female rats. Animals were observed for both narrow eyes and a flattened or elongated nose during the immediate period following treatment on days 1–2. The + sign denotes an observation of one of these signs of pain or discomfort over treatment days 1–2.

	VEH	GEP44	VEH	Exendin–4
**Treatment Day 1**				
**Males**				
**Narrow eyes**		+		+++++++++++
**Flattened/Elongated nose**		+		+++++++++++++++
**Females**				
**Narrow eyes**		++++		+++++++++
**Flattened/Elongated nose**		+++		++++++++
**Treatment Day 2**				
**Males**				
**Narrow eyes**		+++		++++++++++++++
**Flattened/Elongated nose**		+++		+++++++++++
**Females**				
**Narrow eyes**		+++		+++++++++++
**Flattened/Elongated nose**		++++		+++++++++++

*n* = 19/group (males); *n* = 23/group (females).

**Table 2 ijms-26-03032-t002:** Gene expression in IBAT and IWAT following SC vehicle, GEP44 (50 nmol/kg) or exendin-4 (10 nmol/kg) treatment: (**A**) IBAT gene expression following GEP44 and exendin-4 treatment in male HFD-fed rats (*n* = 6–7/group); (**B**) IBAT gene expression following GEP44 and exendin-4 treatment in female HFD-fed rats; (**C**) IWAT gene expression following GEP44 and exendin-4 treatment in male HFD-fed rats (*n* = 7–8/group); and (**D**) IWAT gene expression following GEP44 and exendin-4 treatment in female HFD-fed rats. Animals were fasted for 4 h prior to receiving vehicle, GEP44 or exendin-4 once daily for 3 consecutive days. IBAT and IWAT tissue from male and female HFD-fed rats were collected on treatment day 3 at 2-h post-injection. Data are expressed as mean ± SEM. * *p* < 0.05 GEP44 or exendin-4 vs. vehicle.

**(A** **). Changes in IBAT mRNA Expression Following GEP44 and Exendin-4 Treatment in Male DIO Rats**			
**SC Treatment**	**Vehicle**	**Exendin-4**	**GEP44**
**IBAT**			
*Adrb1*	1.0 ± 0.1	0.4 ± 0.03 *	0.5 ± 0.1 *
*Adrb3*	1.0 ± 0.2	0.9 ± 0.1	0.8 ± 0.1
*Ucp1*	1.0 ± 0.2	0.6 ± 0.1	0.5 ± 0.1
*Cidea*	1.0 ± 0.3	2.2 ± 0.3 *	1.2 ± 0.2
*Dio2*	1.0 ± 0.3	0.8 ± 0.1	0.4 ± 0.1 ^†^
*Gpr120*	1.0 ± 0.3	0.5 ± 0.1	0.3 ± 0.1 *
*Prdm16*	1.0 ± 0.1	1.1 ± 0.1	1.3 ± 0.1
*Ppargc1a*	1.0 ± 0.1	1.4 ± 0.1 *	1.0 ± 0.2
**(B** **). Changes in IBAT mRNA Expression Following GEP44 and Exendin-4 Treatment in Female HFD-Fed Rats**			
**SC Treatment**	**Vehicle**	**Exendin-4**	**GEP44**
**IBAT**			
*Adrb1*	1.0 ± 0.2	0.4 ± 0.04 *	0.4 ± 0.04 *
*Adrb3*	1.0 ± 0.2	0.9 ± 0.2	1.2 ± 0.2
*Ucp1*	1.0 ± 0.2	0.6 ± 0.1 ^†^	0.5 ± 0.1 *
*Cidea*	1.0 ± 0.2	1.5 ± 0.2	1.3 ± 0.3
*Dio2*	1.0 ± 0.1	0.4 ± 0.1 *	0.3 ± 0.1 *
*Gpr120*	1.0 ± 0.2	0.3 ± 0.1 *	0.4 ± 0.1 *
*Prdm16*	1.0 ± 0.1	1.4 ± 0.2	1.5 ± 0.2 ^†^
*Ppargc1a*	1.0 ± 0.2	1.2 ± 0.1	1.1 ± 0.1
**(C** **). Changes in IWAT mRNA Expression Following GEP44 and Exendin-4 Treatment in Male DIO Rats**			
**Treatment**	**Vehicle**	**Exendin-4**	**GEP44**
**IWAT**			
*Adrb1*	1.0 ± 0.1	1.1 ± 0.2	1.3 ± 0.2
*Adrb3*	1.0 ± 0.2	1.0 ± 0.3	1.9 ± 0.5
*Ucp1*	1.0 ± 0.2	2.5 ± 0.9	4.8 ± 1.9
*Cidea*	1.0 ± 0.1	1.1 ± 0.2	0.9 ± 0.1
*Dio2*	1.0 ± 0.2	0.8 ± 0.3	0.7 ± 0.2
*Gpr120*	1.0 ± 0.1	1.5 ± 0.2 *	1.7 ± 0.2 *
*Prdm16*	1.0 ± 0.1	1.2 ± 0.2	1.1 ± 0.1
*Ppargc1a*	1.0 ± 0.1	1.4 ± 0.1†	1.6 ± 0.2 *
**(D** **). Changes in IWAT mRNA Expression Following GEP44 and Exendin-4 Treatment in Female HFD-Fed Rats**			
**Treatment**	**Vehicle**	**Exendin–4**	**GEP44**
**IWAT**			
*Adrb1*	1.0 ± 0.1	0.7 ± 0.1	0.8 ± 0.1
*Adrb3*	1.0 ± 0.3	0.5 ± 0.2	0.6 ± 0.3
*Ucp1*	1.0 ± 0.4	2.4 ± 0.9	1.3 ± 0.5
*Cidea*	1.0 ± 0.4	1.0 ± 0.2	0.9 ± 0.3
*Dio2*	1.0 ± 0.3	0.4 ± 0.1 ^†^	0.4 ± 0.05 *
*Gpr120*	1.0 ± 0.3	1.8 ± 0.3 ^†^	0.9 ± 0.3
*Prdm16*	1.0 ± 0.2	1.0 ± 0.1	0.8 ± 0.1
*Ppargc1a*	1.0 ± 0.4	2.5 ± 2.0	1.6 ± 0.6

IBAT was collected in 6–h fasted rats at 2–h post–injection of vehicle; IWAT was collected in 6–h fasted rats at 2–h post-injection of vehicle; exendin-4 (10 nmol/kg) or GEP44 (50 nmol/kg). * *p* < 0.05 vs. vehicle; ^†^ 0.05 < *p* < 0.1 vs. vehicle; *n* = 7–8/group.

**Table 3 ijms-26-03032-t003:** (**A**,**B**) Effects of the chimeric peptide, GEP44 (50 nmol/kg), and the selective GLP-1R agonist, exendin-4 (10 nmol/kg), on plasma hormones in (**A**) male (*n* = 6–7/group) and (**B**) female HFD-fed rats (*n* = 7–8/group). Animals were fasted for 4 h prior to receiving vehicle, GEP44 or exendin-4 1x daily for 3 consecutive days. Blood was collected on treatment day 3 by tail vein nick (glucose) or cardiac stick (leptin, insulin, FGF-21, glucagon, irisin, adiponectin, total cholesterol, free fatty acids) at 2-h post-injection of VEH, exendin-4, or GEP44. Data are expressed as mean ± SEM. * *p* < 0.05 GEP44 or exendin-4 vs. vehicle.

**(A** **). Plasma Measurements Following SC Administration of GEP44 or Exendin-4 in Male DIO Rats**			
**SC Treatment**	**Vehicle**	**GEP44**	**Exendin-4**
**Leptin (ng/mL)**	70.6 ± 11.2	85.4 ± 10.4	80.1 ± 8.8
**Insulin (ng/mL)**	2.2 ± 1.1	5.3 ± 0.9 *	3.4 ± 0.6
**Glucagon (pmol/L)**	6.3 ± 1.0	5.2 ± 0.8	3.4 ± 0.6 *
**Adiponectin (mg/mL)**	10.9 ± 0.7	5.8 ± 0.6 *	7.3 ± 0.5 *
**Irisin (mg/mL)**	5.4 ± 0.4	7.1 ± 0.6 ^†^	7.4 ± 0.9 ^†^
**FFA (mEq/L)**	0.38 ± 0.2	0.66 ± 0.04	0.75 ± 0.11
**Total Cholesterol (mg/dL)**	89.0 ± 8.9	120 ± 9.6 *	111.8 ± 11.9
**Tail Vein Glucose (mg/dL)**	139.8 ± 7.5	121 ± 6.6 *	115.3 ± 7.4 *
**(B** **). Plasma Measurements Following SC Administration of GEP44 or Exendin-4 in Female High Fat Diet-Fed Rats**			
**SC Treatment**	**Vehicle**	**GEP44**	**Exendin-4**
**Leptin (ng/mL)**	16.3 ± 2.8	20.2 ± 5.5	30.3 ± 8.5
**Insulin (ng/mL)**	0.6 ± 0.04	4.2 ± 1.1 *	2.1 ± 0.5 *
**Glucagon (pmol/L)**	46.6 ± 6.5	37.0 ± 8.0	40.9 ± 4.1
**Adiponectin (mg/mL)**	11.0 ± 0.9	5.8 ± 0.3 *	6.5 ± 0.9 *
**Irisin (mg/mL)**	3.1 ± 0.3	4.6 ± 0.7 ^†^	4.9 ± 0.8 ^†^
**FGF-21 (pg/mL)**	80.9 ± 18.6	440.5 ± 81.9 *	458.2 ± 65.5 *
**FFA (mEq/L)**	0.36 ± 0.06	0.74 ± 0.09 *	0.56 ± 0.07 ^†^
**Total Cholesterol (mg/dL)**	67.1 ± 4.0	85.5 ± 8.3 ^†^	78.6 ± 7.0
**Tail Vein Glucose (mg/dL)**	119.1 ± 6.0	92.5 ± 6.8 *	118.3 ± 8.9

Blood was collected in 6-h fasted rats by tail vein nick (glucose) or cardiac stick (leptin, insulin, glucagon, adiponectin, irisin, FFA, and total cholesterol) at 2-h post-injection of vehicle, exendin–4 (10 nmol/kg) or GEP44 (50 nmol/kg). * *p* < 0.05 vs. vehicle; ^†^ 0.05 < *p* < 0.1 vs. vehicle; *n* = 6–7/group in (**A**); *n* = 7–8/group in (**B**).

**Table 4 ijms-26-03032-t004:** TaqMan probes used in qPCR analysis of IBAT and IWAT.

Gene	Applied Biosystems Assay No.	Probe Context Sequence	Lot Number
*Adrb1*	Rn00824536_s1	GGGTGTTCCGCGAGGCCCAGAAACA	2193241
*Adrb3*	Rn01478698_g1	GCAAGGAGCCTGACTTCTGGAGAAA	1848609
*Ucp 1*	Rn00562126_m1	CTCTTCAGGGAGAGAAACGCCTGCC	2113832
*Cidea*	Rn04181355_m1	GGCCTTGTTAAGGAGTCTGCTGCGG	2113832
*Dio2*	Rn00581867_m1	AGACGCCTACAAACAGGTTAAATTG	2019779
*Gpr120*	Rn01759772_m1	CCAAGATTTTACAGATCACGAAAGC	1959270
*Prdm16*	Rn01516224_m1	GTGAAAACTGCGTCAAGGTGTTCAC	2222767
*Ppargc1a*	Rn00580241_m1	AGACGCCTACAAACAGGTTAAATTG	1984430
*Nono*	Rn01418995_g1	TGGATTGACTCCACCAACAACTGAA	1974129
*Gapdh*	Rn01775763_g1	CCATTGGAGGGCAAGTCTGGTGCCA	2212456

## Data Availability

The original contributions presented in the study are included in the article/Appendix A, further inquiries can be directed to the corresponding author.

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
