# Peer review of "The Chimeric Peptide (GEP44) Reduces Body Weight and Both Energy Intake and Energy Expenditure in Diet-Induced Obese Rats"

_ijms, 2025, doi:10.3390/ijms26073032_

Round 1

Reviewer 1 Report

Comments and Suggestions for Authors

Dear Dr Blevins,

It was a pleasure to read your manuscript. It is well-written, methodologically detailed, and presents valuable findings that could open new avenues in drug discovery for obesity treatment.

Your results are particularly intriguing, as GEP44 did not stimulate energy expenditure (EE), respiratory exchange ratio (RER), core temperature, or activity in rats. Instead, the weight loss effects appear primarily driven by reduced energy intake and increased lipid oxidation rather than enhanced metabolism. This finding raises several important questions:

  • Could GEP44 influence food palatability, thereby reducing intake?
  • Does it affect hypothalamic NPY receptor expression, thereby reducing intake? 
  • Is there a potential link between gut microbiome composition and dietary effects/absorption?

Additionally, I was curious about potential environmental influences. The rats were individually housed, was there access to a running wheel or other environmental enrichment? A comparison to CLAMS, where such enrichment is absent, could provide further insight.

Overall, this is a comprehensive and well-conducted study. I recommend acceptance in its current form and look forward to reading future research from your group.

Author Response

Please see attached comments to Reviewer 1.

We appreciate the generous and helpful feedback from Reviewer 1. 

Reviewer 2 Report

Comments and Suggestions for Authors

Comments to the Authors,

I read with great interest the manuscript entitled “The Chimeric Peptide (GEP44) Reduces Body Weight and Both Energy Intake, and Energy Expenditure in Diet-Induced Obese Rats”.

Major comments:

Methodology:

Was sample size calculated?

The duration of treatment is very short only 3 days, were they enough to produce change?

What about the blood glucose and hormonal measurement, it is written they were assessed once. It would be better to have baseline data and data after and before each intervention.

It would be nice to calculate the percent change with the vehicle, exendin and GEP44 and compare them together by post-hoc analysis.

Results:

It would be nice to add a comparison between males and females.

Figure 3 is missing and figure 4 is in place of figure 3.

It would be nice to propose a guideline for the use of ultrasound in lipodystrophy diagnosis and staging, discuss the barriers to its use.

Limitations:

It would be nice to add a limitations section.

Minor comments:

It would be nice to avoid using the term our and we and replace them with the current study.

Author Response

Please see attached response to Reviewer 2.

We appreciate the generous and helpful feedback from Reviewer 2.
